# Electron-Beam-Induced Grafting of Chitosan onto HDPE/ATZ Composites for Biomedical Applications

**DOI:** 10.3390/polym13224016

**Published:** 2021-11-20

**Authors:** Maria Giulia Faga, Donatella Duraccio, Mattia Di Maro, Christelle Kowandy, Giulio Malucelli, Federico Davide Mussano, Tullio Genova, Xavier Coqueret

**Affiliations:** 1Istituto di Scienze e Tecnologie per l’Energia e la Mobilità Sostenibili (STEMS)-UOS di Torino, Consiglio Nazionale delle Ricerche, Strada delle Cacce 73, 10135 Torino, Italy; mattia.dimaro@unifi.it; 2Institut de Chimie Moléculaire de Reims (ICMR), UMR CNRS 7312, Université de Reims Champagne-Ardenne, Moulin de la Housse, BP 1039, CEDEX 2, 51687 Reims, France; christelle.kowandy@univ-reims.fr (C.K.); xavier.coqueret@univ-reims.fr (X.C.); 3Politecnico di Torino-Dipartimento di Scienza Applicata e Tecnologia, and Local INSTM Unit, Viale Teresa Michel 5, 15121 Alessandria, Italy; giulio.malucelli@polito.it; 4Dipartimento di Scienze Chirurgiche CIR Dental School, Università di Torino, via Nizza 230, 10126 Torino, Italy; federico.mussano@unito.it; 5Dipartimento Scienze della Vita e Biologia dei Sistemi, Università di Torino, via Accademia Albertina 13, 10123 Torino, Italy; tullio.genova@unito.it

**Keywords:** high-density polyethylene (HDPE), alumina-toughened zirconia (ATZ), chitosan, radiation-induced grafting, electron-beam irradiation, wettability, nano-roughness, mechanical properties, cell adhesion, viability

## Abstract

The surface functionalisation of high-density polyethylene (HDPE) and HDPE/alumina-toughened zirconia (ATZ) surfaces with chitosan via electron-beam (EB) irradiation technique was exploited for preparing materials suitable for biomedical purposes. ATR–FTIR analysis and wettability measurements were employed for monitoring the surface changes after both irradiation and chitosan grafting reaction. Interestingly, the presence of ATZ loadings beyond 2 wt% influenced both the EB irradiation process and the chitosan functionalisation reaction, decreasing the oxidation of the surface and the chitosan grafting. The EB irradiation induced an increase in Young’s modulus and a decrease in the elongation at the break of all analysed systems, whereas the tensile strength was not affected in a relevant way. Biological assays indicated that electrostatic interactions between the negative charges of the surface of cell membranes and the –NH_3_^+^ sites on chitosan chains promoted cell adhesion, while some oxidised species produced during the irradiation process are thought to cause a detrimental effect on the cell viability.

## 1. Introduction

Chitosan is the deacetylated polysaccharide from chitin, the natural biopolymer mainly found in shells of crustaceans, as well as in certain fungi cell walls and insects [1,2]. As a natural multifunctional polysaccharide, it has been widely studied for biomedical, pharmaceutical, surgical, and tissue engineering applications [3]. In fact, after deacetylation, chitosan turns into a polycation, due to the protonation of the free amino groups of the D-glucosamine residues, which can easily interact with proteins, lipids, DNA, and, in general, with synthetic polymers negatively charged. This characteristic of chitosan contributes to an increase in solubility, biodegradability/biocompatibility, hemostasis, muco-adhesion, and antimicrobial properties. In addition, chitosan can be considered a low-cost and eco-friendly biopolymer [4].

In bone engineering, both in vitro and in vivo assays have demonstrated that chitosan-based biocomposite scaffolds may favour tissue regeneration [5]. Chitosan is highly biocompatible when used as a wound dressing material [6], an indication already approved by the Food and Drug Administration (FDA). Further, in the human body, no allergic and inflammatory reactions after topical implantation have been observed [3,7]. Some clinical applications of chitosan in jawbone regeneration and alveolar bone have also been reported, showing a reconstruction of critical size defects and an acceleration of dental implant osseointegration [8,9].

Despite all these specific features, the use of chitosan alone in scaffolds is limited by its reduced bioactivities and poor mechanical properties. One of the possible strategies studied to overcome these disadvantages is the blending of the material with other synthetic or natural polymers [10,11,12]. Recently, the functionalisation via radiation-induced grafting has emerged as a promising tool for improving the mechanical properties of natural polymers by the introduction of organic compounds into a polymeric matrix. In some cases, the process is also able to modify the hydrophobic behaviour of synthetic polymers and improve their compatibility with human tissue or biological environments [13]. 

High-density-polyethylene (HDPE) is a highly versatile biomaterial already used in pre-clinical studies and clinical practice, with interesting outcomes [14]. It is characterised by high ductility and ease of processing, though its interaction with cells needs to be improved. Radiation-induced grafting can be an alternative to produce new PE hybrids materials with tunable properties as a function of the specific applications. In the past, for applications outside the biomedical field, this process was used for introducing polymers of acrylic acid [15], methyl methacrylate [16], and acrylamide [17], onto the PE surface via radical pathways using γ-ray or electron beam (EB) irradiation, followed by graft-polymerisation. 

By considering that chitosan cannot be used per se as tissue analogue replacement, in this study, it was grafted onto an HDPE surface after activation via electron-beam irradiation. The aim is to obtain materials for biomedical purposes with improved mechanical and biocompatibility properties with respect to the neat polymer. In a previous study, alumina-toughened zirconia (ATZ) was used as filler with the purpose to endow HDPE with enhanced mechanical properties, hence widening the application field of the polyolefin as biomaterial [18]. Then, the irradiation and grafting of HDPE composites containing ATZ were thoroughly investigated in order to study the influence of this filler on the two processes (i.e., electron-beam irradiation and chitosan grafting reaction). Mechanical properties, wettability, cells adhesion, and viability were investigated. 

## 2. Materials and Methods

### 2.1. Materials 

HDPE (HOSTALEN GF 4750), with melt flow index of 0.4 g/10 min (190 °C/2.16 kg), was kindly supplied by Lyondell–Basel (Kerpen, Germany). Zirconium dioxide reinforced with alumina (ATZ) was supplied by Tosh (Tokyo, Japan); it contains 80 wt% of 3Y-TZP and 20 wt% of alumina. 3Y-TZP consists of 3 mol yttria-stabilised zirconia containing minor components. Before processing, ATZ was ground and dried in a vacuum oven at 80 °C for 2 h. 

Low molecular weight chitosan (deacetylation degree > 75%) was purchased from Sigma Aldrich (Milan, Italy). Before use, it was purified twice by solubilisation in a 1 vol% aqueous acetic acid solution, filtered with filter paper, precipitated with a 1 vol% aqueous sodium hydroxide solution, and washed extensively with deionised water until neutral pH was reached [19].

### 2.2. Composites Preparation 

HDPE composites were prepared as previously described [18]. In brief, HDPE pellets and ATZ powders were mixed in a twin-screw extruder (DSM Xplore Micro 15cc) (Thermo Fisher Scientific Hemel Hempstead, UK) at 190 °C for 5 min. The speed of the extruder screws was set at 60 rpm. The mixing ratios of HDPE/ATZ (wt/wt) were 100/0, 99/1, 98/2, and 93/7. Slabs of HDPE and HDPE/ATZ were obtained by compression moulding in a press Collin P200T at 210 °C for 2 min. The mould had a rectangular shape with dimensions of 100 × 100 mm and a thickness of 0.5 mm. 

### 2.3. Irradiation and Grafting Process

Samples were EB-processed with a low-energy accelerator (Advanced Electron Beam, Dracut, MA, USA) operated at 145 kV so as to ensure a penetration depth of ca. 150 µm. The samples were placed on a conveyor belt passing under the electron beam (1.1 mA current) at a speed of 15 m/min. Under these conditions, the dose for a single pass was 5 kGy. Unfilled HDPE and its composites samples were irradiated by iterative treatments in presence of air to achieve the desired cumulated doses of 50 or 100 kGy. These doses were chosen because, in the literature, it is reported that these values are able to induce significant surface oxidation [20], yet with limited impact on PE properties (mechanical, ageing) [21].

After irradiation, the grafting process was conducted by soaking the irradiated samples into an aqueous acetic acid solution containing chitosan at 1 wt%, under argon atmosphere at a temperature of 70 °C at a pH of 6. For neat HDPE, different reaction times (30 min, 1 h, 2 h, 3 h, 5 h, 7 h, and 15 h) were used in order to find the best grafting conditions. After the grafting reaction, the materials were washed with 1% aqueous acetic acid solution, dried, and stored at −80 °C. For the surface modification of composites HDPE/ATZ 99/1, 98/2, and 93/7, the reaction time was set to 3 h.

### 2.4. Characterisation

#### 2.4.1. FTIR Spectroscopy

FTIR spectroscopy measurements were performed using a PerkinElmer Frontier, (PerkinElmer, Waltham, MA, USA) equipped with an attenuated total reflection (ATR) diamond probe. Spectra (32 scans) of irradiated and chitosan grafted materials were recorded between 4000 and 400 cm^−1^, with a spectral resolution of 4 cm^−1^. All the spectra were normalised with respect to the intensity of the characteristic C-H stretching band of PE at 2916 cm^−1^ [22].

#### 2.4.2. Surface Wettability

Changes in the surface wettability of the irradiated and grafted materials were evaluated by contact angle measurements, performed with a Krüss DSA 100 apparatus equipped with 25× optical zoom (Hamburg, Germany). The sessile drop method was used, and the measurements were taken at room temperature with double distilled water. At least five independent measurements were taken on the surface of each sample.

#### 2.4.3. Nano-Roughness

The surface roughness of irradiated materials was determined by means of atomic force microscopy (AFM), using a Park System XE–100 microscope in contact mode (with Nanosensors PPP-CONTSCR cantilever).

#### 2.4.4. Wide-Angle X-ray Powder Diffraction

The crystallinity degree in irradiated HDPE/ATZ composites was quantified from their wide-angle X-ray powder diffractograms (WAXD), using a PW3040/60 X'Pert PRO MPD diffractometer from PANalytical in Bragg–Brentano geometry (Malvern, UK). WAXD profiles were acquired using a Ni-filtered Cu Kα radiation (λ = 0.15418 nm), working at 45 kV and 40 mA with a continuous scan of 0.04°/s (scan step size: 0.0167°; time per step: 53 s) in the range of 5–70°. The crystallinity index *x_c_* was calculated as the ratio between the intensity of the crystalline phase diffraction and the intensity of the total sample diffraction, according to Equation (1).
(1)$$x_{c}=\frac{I_{c\left(PE\right)}}{I_{TOT}-I_{cATZ}}\times 100$$
where *I_c(PE)_ = I_TOT_* – *I_AM(PE)_* and $I_{TOT}=I_{AM\left(PE\right)}+I_{c\left(PE\right)}+I_{cATZ}$.

#### 2.4.5. Tensile Properties

The tensile properties of the irradiated samples were measured using an Instron 5566 tensile tester (West Sussex, UK) on the compression-moulded plats (ASTM D882) at room temperature and 50% R.H. The dimensions of the specimens were 0.5 mm of thickness and 5 mm of width. The parameters at break (elongation (ε_b_) and strain (σ_b_)) were evaluated using a constant deformation rate, with the aim to keep the ratio v/L_0_ equal to 10 mm/(mm·min); where v is the deformation rate, and L_0_ is the initial length of the specimen. Young’s modulus E was measured at a constant speed, given from the ratio v/L_0_ = 0.1 mm/(mm·min). At least five independent measures were carried out on each sample. 

#### 2.4.6. Biological Assays 

Biological assays in vitro were conducted with the fibroblast cell line NHDF (ECACC, Salisbury, UK). As reported in a previous study by our group [23], cells were maintained at 37 °C in Dulbecco’s modified Eagle medium (DMEM), supplemented with 10 vol% foetal bovine serum (Life Technologies, Milan, Italy), 100 U/mL penicillin, and 100 μg/mL streptomycin, under a humidified atmosphere of 5 vol% CO_2_ in air. To evaluate cell adhesion on grafts, cells were detached using trypsin for 3 min, carefully counted, and seeded at 2 × 10^3^ cells/disk in 100 μL of growth medium on the samples, in 24 well plates. After being kept for 10 min in an incubator, the grafts were washed with phosphate-buffered solution (PBS) and immediately stained with DAPI (4′,6-diamidino-2-phenylindole dihydrochloride) to mark the nuclei. Thus, the number of DAPI-positive nuclei enabled the determination of the adherent cells. To analyse cell morphology, fibroblasts were plated at a density of 2500 cells/sample in 24-well and kept in an incubator. Cell viability (at 24 h) was assessed by Cell Titre GLO (Promega, Milan, Italy).

## 3. Results

### 3.1. Grafting by Radiation-Induced Peroxidation of HDPE-Based Substrates

To immobilise chitosan at the surface of HDPE-based materials, we selected a radiation-induced peroxidation method which seemed well adapted to the objective of producing, by simple and scalable procedures, surface-modified composites with enhanced properties for biomedical applications.

The pre-irradiation technique is a well-established method for achieving efficient radiation-induced graft polymerisation onto a variety of synthetic or natural organic polymers [24,25]. A less common approach consists of performing a coupling reaction between a pre-activated material and a second polymer to be immobilised on the substrate.

Irradiation of HDPE in solid-state and under inert conditions at doses up to some tens kGy is known to induce the radiolysis of the hydrocarbon chains with the generation of C-centred free radicals, the evolution of H_2_, formation of ethylenic unsaturations, interchain linkages, and few chain scissions [26]. Due to mobility restrictions inherent to polyolefin chains and segments forming the crystalline domains, the effects of radiolysis are essentially limited to the formation of C-centred radicals and hydrogen evolution, whereas the sequence of reactions leading to the formation of new functional groups occurs in the amorphous domains which are in the rubbery state at room temperature. When irradiation is conducted in air, the radiolytic process and the subsequent reactions that occur in the amorphous domains are affected by the presence of O_2_, which interacts with the formed free radicals [27,28]. This reduces to some extent the radiation-induced structural modifications listed above, to the benefit of the formation of peroxyl radicals which are further converted into hydroperoxides, peroxides, and various types of carbonyl functions [29]. The relative amount of the formed products is dependent on the conditions of treatment and on the conditions and duration of storage for the irradiated substrate.

Chitosan, similar to most of the other polysaccharides, is sensitive to hydrogen abstraction induced by a variety of oxidising free radicals, particularly the hydroxyl radical HO^•^ [30]. We, therefore, envisioned the presented method based on the reaction between the peroxidised PE substrate and chitosan solutions under appropriate conditions.

The efficiency of the grafting stage is expected to be dependent (i) on the number and decomposition rate of the peroxyl groups, the latter being activated by temperature and by optional redox additives such as ferrous ions; (ii) on the reactivity and concentration of the polymer to be grafted at the liquid–solid interface between the substrate and the polymer solution. In a simplified description of the grafting stage of this method, it is expected that the decomposition of hydroperoxides induces the formation of radicals HO^•^ which diffuse in the vicinity of the substrate surface and abstract labile hydrogen atoms from the glucosamine units. The recombination of the resulting C-centred free radicals present on chitosan chains with some alkoxyl radicals on the substrate should result in the covalent coupling between the two distinct polymers, hence inducing chitosan immobilisation. Alternatively, the addition of C-centred free radicals on C=C unsaturations of the HDPE would also link the two polymers, but the kinetic chain would not be stopped with this type of reaction.

The following scheme gives a sequential representation of the reactions and transformations occurring at the various stage of the process (Figure 1).

The major products formed upon EB irradiation of polyethylene have characteristic infrared absorption bands [31]. Figure 1A–E show five main regions of ATR spectra for HDPE and its composites before and after EB irradiation at 100 kGy. For sake of simplicity, the spectra of the samples irradiated at 50 kGy are not reported, as no significant differences can be observed with respect to the counterparts irradiated at 100 kGy. All the spectra have been normalised with respect to the intensity of the characteristic C–H stretching band of PE at 2916 cm^−1^ [22]. Regarding the FTIR spectrum of HDPE, the fundamental stretching and bending vibration bands appear and correspond well with the literature [32,33]. After irradiation, a more evident and pronounced peak at 3398 cm^−1^ appears (Figure 1A), which corresponds to the O–H vibration mode. Further, a new peak centred at 1715 cm^−1^ due to C=O stretching mode is visible in Figure 1B. These findings confirm the occurrence of oxidation during EB irradiation. However, it is not possible to differentiate the contribution of hydroperoxide species and possibly formed hydroxy groups to the overall O–H vibration mode assigned to the band centred at 3500 cm^−1^ [34]. A weak absorption band at 968 cm^−1^ is also visible (Figure 1D): it can confidently be assigned to the formation of the trans-vinylene (–CH=CH–) group, suggesting the gradual formation of unsaturations on the PE chains, as discussed previously (Figure 1) [35]. Furthermore, after irradiation, the absorption bands at 1470 (δa–CH_2_ bending fundamental, Figure 1C) and 718 (γr–CH_2_ rocking fundamental, Figure 1D) cm^−1^ are split into two bands. These fundamental modes appear as doublets due to correlation splitting associated with new inter-chain interactions in the crystalline regions of PE as a consequence of the irradiation process, thus resulting in the identification of the B1u (higher frequency) and the B2u (lower frequency) species for each normal mode [34]. 

As far as the HDPE/ATZ composites are concerned, independently of the irradiation dose and the ATZ loading, they behave similarly to the unfilled polymer. However, the intensities of the absorption bands at 1715 and 3400 cm^−1^ (which correspond to C=O and O–H stretching modes, respectively) decrease with increasing the ATZ loading. This finding can be ascribed to a possible reduction of oxide particles under electron-beam irradiation [36], which results in protection of the polymer, preventing formation and recombination of radical species. Thus, the presence of C=O and O–H groups lowers. Moreover, there are also other possible explanations for this feature—namely, (i) the larger the content is in ATZ, the lower is the effective dose interacting in the probed depth of the irradiated samples, hence reducing the degradation process; (ii) the change in refractive index and ATR penetration.

As reported in the literature, EB irradiation changes the chemical structure, as well as the surface nano-morphology up to a few nano-metres in depth [37,38]. As shown in Table 1, the nano-roughness, measured by AFM, increases in all the irradiated samples with respect to the untreated counterparts. However, the presence of ATZ seems to reduce the effect of the EB on the nano-roughness. In fact, the roughness of HDPE after irradiation increases by about 107%, while in HDPE/ATZ 93/7 composites, it increases only by about 12%. The result is consistent with the fact that ATZ particles interact with electrons, preventing further reactions [36], such as polymer scavenging, hence leading to a higher roughness. 

The wettability of HDPE and its composites (Table 1) changes significantly after irradiation; moreover, all the composites show the same behaviour. This means that surface nano-roughness is not the only parameter affecting wettability, in agreement with our previous outcomes [11,23,39]. In this case, the high increase in roughness (almost doubled) for unfilled HDPE and its composite with only 1 wt% of ATZ can be responsible for a significant wettability enhancement. In the case of high ATZ loadings, the most significant changes induced by irradiation are related to crystallinity enhancement and probable surface charge, which both modify the wettability. 

Table 1 also reports the crystallinity degree (*x_c_*) of the irradiated materials. According to the literature [40], *x_c_* increases after the irradiation because of the crystallisation of tie chain molecules that undergo scission in the amorphous region during the process. The crystallinity increase is more relevant for the HDPE/ATZ 93/7 composite. Additionally, this finding confirms the ATZ's ability to trap charges, acting as a nucleating agent for crystallisation [41]. 

All the tensile parameters of irradiated materials both at 50 and 100 kGy, obtained by stress–strain tests, are collected in Table 2. The data of untreated HDPE and its composites, already gathered in a previous study [18], are reported for comparison. From a general point of view, the irradiation process induces an increase in Young’s modulus and a decrease in the elongation at break, whereas the tensile strength is not significantly affected. Similar behaviour was found also for irradiated polymer materials [42]. 

The radiation dose does not seem to affect the mechanical behaviour of the investigated systems. In particular, the increase in Young’s modulus for HDPE/ATZ 93/7 composite is similar to that of HDPE and other composites; therefore, it does not reflect the difference in crystallinity among these materials. In fact, Young’s modulus increases by 16% and x_c_ by 13% for HDPE/ATZ 93/7, compared with Young’s modulus 15–18% and x_c_ +3–4% for HDPE/ATZ 99/1 and neat HDPE. This finding could be probably ascribed to the higher number of aggregates and voids present in this composite [18], which counterbalance the higher crystallinity induced in the matrix by EB treatment. 

### 3.2. Chitosan Grafting Reaction

The presence of chitosan grafted onto the HDPE surface would lead to a change in the chemical composition of the surface and to an increase in polymer wettability. For this reason, the chitosan functionalisation at different reaction times was monitored by ATR–FTIR spectroscopy and contact angle measurements. 

Figure 2A–F show the ATR–FTIR spectra of HDPE surfaces after different grafting reaction times. Specifically, Figure 2A–C refer to the three ATR regions, according to which it is possible to visualise chitosan grafted onto HDPE surface activated by EB irradiation at 50 kGy. Figure 2D–F report the spectra of HDPE surfaces activated at 100 kGy. In some cases, bands of chitosan are overlapped with the existing ones due to the irradiation process. The band due to the vibrational mode of the OH group (νO–H∼3400 cm^−1^) appears within 4000 and 3000 cm^−1^ (Figure 2A,D). In grafted HDPE, this band is more intense with respect to that of the ungrafted counterpart. Two bands appearing in the region from 1800 to 1500 cm^−1^ (Figure 2B,E) are attributable to the protonated amine group—namely, the anti-symmetrical deformation that eventually overlapped the amide I band of the N-acetylated glucosamine units at ~1650 cm^−1^, and the symmetric deformation (δ_NH3_) at ~1560 cm^−1^ [43,44]. Furthermore, Figure 2C,F show the vibrational bands of the pyranose ring at ∼ 1080 cm^−1^, confirming the presence of chitosan onto the HDPE surface.

Interestingly, all these bands characteristic to chitosan chemical changes increase in intensity with increasing the reaction time, reaching a maximum after 3 h and then decreasing for longer reaction time. Relevant differences cannot be found among the irradiated samples at 50 and 100 kGy, except for the fact that the bands appear more intense in the sample irradiated at 100 kGy.

This behaviour is also confirmed by the wettability measurement. In fact, the contact angle values (Figure 3A,B) decrease by increasing reaction time and achieving a minimum value after three hours of reaction. Then, the contact angles slightly increase again, reaching a plateau at higher reaction times. 

As grafting is expected to occur gradually with the continuous progress of the peroxide decomposition and subsequent coupling of chitosan, the surface effects observed by contact angle measurements are expected to appear similarly, in a monotonous way, likely with some saturation and levelling-off for the wettability by water [45]. We actually observe an inversion point, suggesting the occurrence of a more complex process.

Two phenomena can be responsible for the increase in the contact angles after 3 h of reaction—namely, (i) a considerable amount of chitosan already grafted onto HDPE backbone creates a steric hindrance for further grafting, or (ii) some degradative chain transfer in the radical process or bond scission at some hydrolytically unstable functions could cause the debonding or reduction in length of the chitosan chains attached at the surface of the HDPE substrate. The degradation of chitosan in an aqueous solution by hydrogen peroxide and peracids is indeed a well-established process [46]. A rough representation of this degradation process which competes with the grafting is provided in Figure 2a. We cannot exclude the possibility of a hydrolytic scission; however, as the reaction is based on the in situ decomposition of hydroperoxides generated during the pre-irradiation stage, we consider it very likely to have in the medium some reactions involving HO• radicals just released at the surface and of immobilised chitosan chains. The degradation of chitosan in an aqueous solution by hydrogen peroxide and peracids is indeed a well-established process. This tentative explanation is represented in Figure 2b, where one of the possible sites of a glucosamine unit is subject to an H abstraction. The unstable free radical formed at the C4 position of the pyranose ring would undergo a rearrangement with scission of the polysaccharide chain and the release of a ketone end-group, on the one hand, whereas the second segment would be released with C-centred free radical readily reacting with molecular oxygen in the aerated medium, on the other. 

We aim to conduct a deeper investigation on this phenomenon, including the quantification of peroxides present at the surface of the substrate, assessing the amount of immobilised chitosan as a function of grafting conditions and duration, and measuring the probable release of chitosan segments from the grafted substrates in representative conditions of the grafting procedure. 

Based on the results obtained for neat HDPE, the grafting reaction for the composites was carried out for 3 h after EB irradiation at the dose of 100 kGy. The ATR–FTIR spectra in different regions of infrared absorption bands and contact angle values of the composites before and after 3 h of grafting reaction are depicted in Figure 4 and Table 3, respectively. 

After functionalisation, the HDPE/ATZ 99/1 composite (line b) shows increased absorption bands with respect to the ungrafted counterpart (line a): this finding can be ascribed to the presence of chitosan (bands at 3398, 1650–1560, and 1080 cm^−1^). The same bands in the HDPE/ATZ 98/2 and 97/3 composites do not increase so significantly, hence indicating that, for these composites, the functionalisation with chitosan becomes less evident. The contact angle values are in agreement with the ATR observations. In fact, for unfilled HDPE and HDPE/ATZ 99/1 composites, θ decreases to 66°; for HDPE/ATZ 98/2 and 93/7 composites, it remains almost constant, hence indicating a low percentage of grafted chitosan onto the surfaces. The presence of ATZ exceeding 2 wt% hinders both the EB irradiation process and the chitosan functionalisation. 

Biological assays in vitro were performed with fibroblast cells on HDPE and HDPE/ATZ composites functionalised with chitosan (reaction time: 3 h) via EB irradiation (dose of 100 kGy). Figure 5 shows cell adhesion at 10 min (A) and cell viability after 24 h (B) of untreated and chitosan-functionalised HDPE and HDPE/ATZ composites. All the composites containing ATZ particles sustain a higher number of adherent cells (Figure 5A), with the exception of 93/7, and increase cell viability (Figure 5B), as compared with unfilled HDPE. In particular, cell viability on the 98/2 resulted in the highest of all the samples in a statistically significant way. ATZ may drive the cell response and the simple attraction toward the surface, suggesting that the introduction of a polar phase (ATZ) into a non-polar matrix (HDPE) is sufficient to guarantee an interaction, and thus adhesion and viability. When the ATZ content is only 1 wt%, the effect on cell interaction is negligible, while it becomes significant when the amount is doubled. For higher oxide concentration (7 wt%) some agglomerates appear [18], reducing the number of active sites exposed on the surface, then minimising the ATZ effect on the cell interaction. Similar behaviour was also found in a previous study [23], in which the best dispersion was the prevalent phenomenon on the ATZ surface reactivity. Upon these premises, further inquiries should address the possible role of the dispersion of ATZ within the polymeric matrix as a key factor leading to the cell interaction with the substrate. 

Interestingly, the presence of chitosan-functionalised surfaces further increases cell adhesion (Figure 5A). However, cell viability after 24 h (Figure 5B) significantly decreases. This finding is consistent with the hypothesis of electrostatic interactions occurring between the negative charges of the surface of cell membranes and the –NH_3_^+^ sites on chitosan chains, which first promote interaction, as already reported in the literature [11,47,48,49]. For longer reaction times, the possible presence of unreacted activated oxygen species, such as hydrogen peroxides, on the material surface may have a detrimental effect on the cell viability of fibroblast [50]. Further studies are necessary for gaining a deeper insight into this negative effect. We are considering the possible release of oxidative species that may originate from the excess of hydrocarbon peroxides and hydroperoxides that were left unreacted in our protocol optimised for the grafting stage, to some chemically modified glucosamine units formed as by-products of the grafting stage, or to the slow but long-lasting diffusion of C-centred free radicals trapped into the crystalline domains, which may fix oxygen and generate, at a small but continuous rate, ROS in the culture medium. From a general point of view, it is worthy to conclude that the main effect on cell interaction is mainly given by the chemistry of chitosan and/or surface material and is strictly dependent on roughness and wettability [11,23,39]. 

## 4. Conclusions

In this study, chitosan was grafted onto HDPE and HDPE/ATZ composite surfaces after their activation via EB irradiation, with the aim to obtain biomaterials a new system endowed with enhanced good mechanical properties and high biocompatibility, compared with the neat polymer and its composites. The EB irradiation induced an increase in Young’s modulus and a decrease in the ductility of all analysed systems, whereas the tensile strength was not remarkably affected. In addition, the EB treatment was responsible for the modification, in a different way, of wettability, crystallinity, and nano-roughness of HDPE and its composites. 

Interestingly, the presence of ATZ in amounts equal to or higher than 2 wt% of ATZ (that is able to trap charges) influenced both the EB irradiation process and the chitosan functionalisation reaction, reducing the oxidation sites and the presence of grafted chitosan on the surface. Biological assays indicated that the electrostatic interactions occurring between the negative charges of the surface of cell membranes and the –NH_3_^+^ sites on chitosan chains were able to promote cell adhesion, whereas the surface oxidation due to the irradiation process could induce some detrimental effects on cell viability.

## Data Availability

The data presented in this study are available on request from the corresponding author.

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
