# Peer review of "Electron-Beam-Induced Grafting of Chitosan onto HDPE/ATZ Composites for Biomedical Applications"

_polymers, 2021, doi:10.3390/polym13224016_

Round 1
Reviewer 1 Report
Dear Authors, the manuscript entitled Electron beam-induced grafting of chitosan onto HDPE/ATZ composites for biomedical applications is interesting and has some novelty impact. However before publication in Polymers Journal some elements of manuscript need correction or clarification.
Comments;
- Line 70-82; Authors clearly presented an aim of study. However some results and methods used are reported in this part of manuscript that is not necessary.
- Line 111; Authors wrote ….desired cumulated doses of 50 or 100 kGy. The reason of choice the irradiation condition should be clearly explained or documented by results from optimization.
- Line 121; The clear subsections for each used method for material characterization should be added.
- The quality of scheme 1 and 2 (resolution ) is low and should be improved.
- The y axis description at figure 1 and 2 need explanations in materials and methods. The spectra were normalized by dividing by transmittance value for band at 2916 cm-1 ? The x axis should be described, e.g Wavenumber (cm-1)
- The results presented at figure 2 could be also analyzed by two-dimensional correlation spectroscopy method that allows for quick and reliable analysis of changes and relation between bands in analyzed spectra. The OriginPro software or Mat2dcorr is suitable to perform this analysis. Moreover the values of normalized transmittance (or absorbance) for marked bands could be plotted vs time for better visualization of described changes against showing the ATR-FTIR spectra.
- The table 1 and 2 formatting should be corrected with reduction number of vertical and horizontal lines.
- Line 282; the repetition of “does” should be removed.
- Line 387; Authors wrote; Figure 5 shows cell adhesion… there is not a figure 5 in manuscript
- Line 389; Authors wrote; faster cell adhesion….. but authors checked adhesion only after 10 minutes ?
- On figure 6 are visible a difference between adhesion and viability of fibroblast cells depend on tested composites. However any statistical analysis is not visible that could confirm significance reported differences ?
- How authors explain so high increase in cell viability for 98/2 composite ?
Author Response
We would like to thank the Reviewer for the comments that help the improvement of the manuscript. Below the answers to the comments.
Line 70-82; Authors clearly presented an aim of study. However some results and methods used are reported in this part of manuscript that is not necessary.
We have reduced the paragraph leaving only fundamental comments:
“By considering that chitosan cannot be used per se as tissue analogue replacement, in this work it was grafted onto HDPE surface after activation via electron beam irradiation. The aim is to obtain materials for biomedical purposes with improved mechanical and biocompatibility properties with respect to the neat polymer. In a previous work, Alumina-Toughened Zirconia (ATZ) was used as filler with the purpose to endow HDPE with enhanced mechanical properties, hence widening the application field of the polyolefin as biomaterial [18]. Then, the irradiation and grafting of HDPE composites containing ATZ was thoroughly investigated in order to study the influence of this filler on the two processes (i.e. electron beam irradiation and chitosan grafting reaction). Mechanical properties, wettability, cells adhesion and viability were investigated.”
Line 111; Authors wrote ….desired cumulated doses of 50 or 100 kGy. The reason of choice the irradiation condition should be clearly explained or documented by results from optimization.
The cumulated doses of 50 or 100 kGy have been chosen because in literature it is reported that these values are able to induce significant surface oxidation [Effects of electron-beam irradiation on surface oxidation of polymer composites M. Zenkiewicz, M. Rauchfleisz, J. Czupryn ́ska, J. Polan ́ski, T. Karasiewicz, W. Engelgard. Applied Surface Science 253 (2007) 8992–8999], yet with limited impact on PE properties (mechanical, ageing) [Mizera, M. Manas, Z. Holik, D. Manas, M. Stanek, J. Cerny, M. Bednarik, and M. Ovsik. Properties of HDPE after Radiation Cross-linking. International journal of mathematics and computers in simulation 2012-16-597]. In the paper this comment has been added.
Line 121; The clear subsections for each used method for material characterization should be added.
A subsection for each method has been introduced accordingly.
The quality of scheme 1 and 2 (resolution) is low and should be improved.
We have improved the resolution of scheme 1 and 2.
The y axis description at figure 1 and 2 need explanations in materials and methods. The spectra were normalized by dividing by transmittance value for band at 2916 cm-1 ? The x axis should be described, e.g Wavenumber (cm-1)
The x axis description was reported, adding “wavenumber”, according to reviewer suggestion.
The results presented at figure 2 could be also analyzed by two-dimensional correlation spectroscopy method that allows for quick and reliable analysis of changes and relation between bands in analyzed spectra. The OriginPro software or Mat2dcorr is suitable to perform this analysis. Moreover the values of normalized transmittance (or absorbance) for marked bands could be plotted vs time for better visualization of described changes against showing the ATR-FTIR spectra.
The suggestion to apply two-dimensional correlation spectroscopy with the purpose to get information on systematic correlation among spectroscopic features is really appreciated, since it can provide more insights about the systems under study. Also in the present case it could be useful, although the authors think that a deeper FTIR analysis is out of the purpose of the present investigation. Indeed, spectra reported in Figure 2 provides evidence of the highest intense peaks related to chitosan in the case of sample “e” in all the spectral range (Figure 2 d, e and f). This spectral evidence indicates 3 hours as the most suitable condition for grafting chitosan on the materials surface after irradiation under 100kGy.
The table 1 and 2 formatting should be corrected with reduction number of vertical and horizontal lines.
In table 2 we have removed a line in which irradiation doses was repeated. Table 1 has the minimum number of possible lines.
Line 282; the repetition of “does” should be removed.
In the line the word does is not reported twice. The irradiation dose does not seem to affect the mechanical behavior of the investigated systems.
Line 387; Authors wrote; Figure 5 shows cell adhesion… there is not a figure 5 in manuscript
Figure 5 is Figure 6. This has been corrected.
Line 389; Authors wrote; faster cell adhesion….. but authors checked adhesion only after 10 minutes ?
We agree and amended the manuscript accordingly, pointing out only the different number of adherent cells.
On figure 6 are visible a difference between adhesion and viability of fibroblast cells depend on tested composites. However any statistical analysis is not visible that could confirm significance reported differences ?
We thank the reviewer and added the statistical analysis. 98/2 composite was indeed capable of increasing significantly cell viability.
How authors explain so high increase in cell viability for 98/2 composite ?
It is difficult to hypothesize how this could happen. Further experiments beyond the scope of the present paper ought to be performed to address this point. The text was hopefully improved to explain better this point.
“Figure 5 shows cell adhesion at 10 min (A) and cell viability after 24 h (B) of untreated and chitosan-functionalized HDPE and HDPE/ATZ composites. All the composites containing ATZ particles sustain a higher number of adherent cells (Figure 5A), with the exception of 93/7, and increase cell viability (Figure 5B) as compared to unfilled HDPE. In particular, cell viability on the 98/2 resulted the highest of all the samples in a statistically significant way. ATZ may drive the cell response and the simple attraction toward the surface, suggesting that the introduction of a polar phase (ATZ) into a non-polar matrix (HDPE) is sufficient to guarantee an interaction and thus adhesion and viability. When the ATZ content is only 1wt%, the effect on cell interaction is negligible, while becomes significant when the amount is doubled. For higher oxide con-centration (7wt%) some agglomerates appear [18], reducing the amount of active sites exposed on the surface, then minimizing the ATZ effect on the cell interaction. A simi-lar behaviour was also found in a previous work, where the best dispersion was the prevalent phenomenon on the ATZ surface reactivity. Upon these premises, further inquiries should be addressed on the possible role of the dispersion of ATZ within the polymeric matrix as a key factor leading the cell interaction with the substrate.“
Reviewer 2 Report
Dear authors,
please, the manuscript „Electron beam-induced grafting of chitosan onto HDPE/ATZ composites for biomedical applications“ is interesting because it describes the preparation of various HDPE/ATZ composites with grafted chitosan, and these materials seem to be very attractive for biomedical applications. The paper is of interest, but it needs minor revisions. You can find my recommendation below:
Have the names of figures uniform (with or without space between number and letter, e.g. 2B, 3 B, etc.). Correct typing errors (Scheme 2.a, Scheme 2.b), correct units cm-1, “-1” should be in superscript.
- p.2, l. 72 – What means “new product with good mechanical properties and higher biocompatibility”? Be more specific (what product, properties etc.). Clearly state the aims of your work.
- p. 6, l. 254-255 – The hypothesis “The result is consistent with the fact that ATZ…” needs a reference.
- p. 10, l. 319-322 – This claim needs a reference.
- p. 13 – Correct figure caption. I suppose that it should be Figure 5 instead of Figure 6.
- p. 13-14. Improve discussion of results presented in Figure 6 (or Figure 5). How do you explain differences in cell adhesion and cell viability between tested composites (grafted, non-grafted, various compositions 99/1, 98/2, 97/3)?
- p. 14 – Improve the Conclusions with regard to the general claim “to obtain a new system with good mechanical properties and high biocompatibility”. Thus, the aims of your work will be more obvious to readers.
Author Response
We would like to thank the Reviewer for the comments that help the improvement of the manuscript. Below the answers to the comments.
Dear authors,
please, the manuscript „Electron beam-induced grafting of chitosan onto HDPE/ATZ composites for biomedical applications“ is interesting because it describes the preparation of various HDPE/ATZ composites with grafted chitosan, and these materials seem to be very attractive for biomedical applications. The paper is of interest, but it needs minor revisions. You can find my recommendation below:
Have the names of figures uniform (with or without space between number and letter, e.g. 2B, 3 B, etc.). Correct typing errors (Scheme 2.a, Scheme 2.b), correct units cm-1, “-1” should be in superscript.
Corrections have been done.
p.2, l. 72 – What means “new product with good mechanical properties and higher biocompatibility”? Be more specific (what product, properties etc.). Clearly state the aims of your work.
The text was revised according both reviewers suggestions:
“The aim is to obtain materials for biomedical purposes with improved mechanical and biocompatibility properties with respect to the neat polymer. In a previous work, Alumina-Toughened Zirconia (ATZ) was used as filler with the purpose to endow HDPE with enhanced mechanical properties, hence widening the application field of the polyolefin as biomaterial [18]. Then, the irradiation and grafting of HDPE composites containing ATZ was thoroughly investigated in order to study the influence of this filler on the two processes (i.e. electron beam irradiation and chitosan grafting reaction). Mechanical properties, wettability, cells adhesion and viability were investigated.”
- 6, l. 254-255 – The hypothesis “The result is consistent with the fact that ATZ…” needs a reference.
A reference has been added
- 10, l. 319-322 – This claim needs a reference.
We have added a reference and revised the sentence
As grafting is expected to take place gradually with a continuous progress of the peroxide decomposition and subsequent coupling of chitosan, the surface effects observed by contact angle measurements are expected to appear similarly, in a monototonous way, likely with some saturation and levelling-off for the wettability by water [Evidence that three-regime kinetics is inherent to formation of a polymer brush by a grafting-to Approach Xue Sha, Xiaohe Xu, Karl Sohlberg, Patrick J. Lollb and Lynn S. Penn. RSC Adv., 2014, 4, 42122]. We actually observe an inversion point suggesting the occurrence of more complex process.
- 13 – Correct figure caption. I suppose that it should be Figure 5 instead of Figure 6.
Figure caption has been corrected
- 13-14. Improve discussion of results presented in Figure 6 (or Figure 5). How do you explain differences in cell adhesion and cell viability between tested composites (grafted, non-grafted, various compositions 99/1, 98/2, 93/7)?
We have improved the discussion as here reported:
“Figure 5 shows cell adhesion at 10 min (A) and cell viability after 24 h (B) of untreated and chitosan-functionalized HDPE and HDPE/ATZ composites. All the composites containing ATZ particles sustain a higher number of adherent cells (Figure 5A), with the exception of 93/7, and increase cell viability (Figure 5B) as compared to unfilled HDPE. In particular, cell viability on the 98/2 resulted the highest of all the samples in a statistically significant way. ATZ may drive the cell response and the simple attraction toward the surface, suggesting that the introduction of a polar phase (ATZ) into a non-polar matrix (HDPE) is sufficient to guarantee an interaction and thus adhesion and viability. When the ATZ content is only 1wt%, the effect on cell interaction is negligible, while becomes significant when the amount is doubled. For higher oxide con-centration (7wt%) some agglomerates appear [18], reducing the amount of active sites exposed on the surface, then minimizing the ATZ effect on the cell interaction. A simi-lar behaviour was also found in a previous work, where the best dispersion was the prevalent phenomenon on the ATZ surface reactivity. Upon these premises, further inquiries should be addressed on the possible role of the dispersion of ATZ within the polymeric matrix as a key factor leading the cell interaction with the substrate. “
- 14 – Improve the Conclusions with regard to the general claim “to obtain a new system with good mechanical properties and high biocompatibility”. Thus, the aims of your work will be more obvious to readers.
According to the reviewer suggestions, the general claim in the Conclusion has been changed as follows:
“In this work, chitosan was grafted onto HDPE and HDPE/ATZ composite surfaces after their activation via electron beam irradiation, with the aim to obtain biomaterials endowed with enhanced good mechanical properties and biocompatibility with respect to the neat polymer and its composites. “
Reviewer 3 Report
The review manuscript “Electron beam-induced grafting of chitosan onto HDPE/ATZ composites for biomedical applications” by Maria Giulia Faga, et al. investigated the surface functionalization of High-Density Polyethylene (HDPE) and HDPE/Alumina 18 Toughened Zirconia (ATZ) with chitosan via electron beam (EB)-irradiation. Their finding suggested that the EB irradiation induced an increase of the Young’s modulus and a decrease of elongation. They also subjected these materials to biological assessment and their results indicated that electrostatic interactions between the negative charges of the surface of cell membranes and the NH3+ sites on chitosan promoted cell adhesion, while some oxidized species produced during the irradiation possibly caused a detrimental effect on the cell viability.
I found the overall study design appropriate for the purpose of this investigation with a need for few additional experiments:
- Given the ATR-FTIR characterization of chemical changes in HDPE and composites captured for different regions of infrared absorption bands, it would be valuable to use thermoanalytical assessment like DSC to provide more inside on structural thermodynamics of these changes.
- To show significant changes of surface roughness due to irradiation, in addition to contact angle measurement and assessing wettability, it is crucial to look at surface nanotopography using AFM or SEM to better capture those changes at the nano- and micro- scales.
- Given biological assessment data using fibroblast cells, it would be crucial to check if temperature aging can affect this outcome as part of their materials shelf life. Additionally, based on 24 h significant decrease in viability using Cell 165 Titer GLO assay which is based on ATP quantification, and not fully revealing the cytotoxicity of these surfaces, it would be crucial to couple this biological assessment with at least another assay like LDH cytotoxicity to evaluate the presence any cell damage because of this exposure.
Author Response
Reviewer 3
The review manuscript “Electron beam-induced grafting of chitosan onto HDPE/ATZ composites for biomedical applications” by Maria Giulia Faga, et al. investigated the surface functionalization of High-Density Polyethylene (HDPE) and HDPE/Alumina 18 Toughened Zirconia (ATZ) with chitosan via electron beam (EB)-irradiation. Their finding suggested that the EB irradiation induced an increase of the Young’s modulus and a decrease of elongation. They also subjected these materials to biological assessment and their results indicated that electrostatic interactions between the negative charges of the surface of cell membranes and the NH3+ sites on chitosan promoted cell adhesion, while some oxidized species produced during the irradiation possibly caused a detrimental effect on the cell viability.
I found the overall study design appropriate for the purpose of this investigation with a need for few additional experiments:
1. Given the ATR-FTIR characterization of chemical changes in HDPE and composites captured for different regions of infrared absorption bands, it would be valuable to use thermoanalytical assessment like DSC to provide more inside on structural thermodynamics of these changes.
We think that DSC cannot provide more insight on the changes observed by ATR. Indeed, ATR account for changes observed in few microns (no more than 2) and not information on the deeper layers can be obtained this way. The effect of EB on bulk properties is studied by other measurements: i) crystallinity, measured by XRD, more reliable than DSC for this property; ii) mechanical behaviour, obtained by tensile tests.
2. To show significant changes of surface roughness due to irradiation, in addition to contact angle measurement and assessing wettability, it is crucial to look at surface nanotopography using AFM or SEM to better capture those changes at the nano- and micro- scales.
Thank you for the suggestion, useful to usually better explain the differences. In the attached file some representative AFM images are reported for showing the nanotopography before and after the irradiation treatments. Unfortunately, the results do not allow to obtain significant information about the nanotopography changes.
3. Given biological assessment data using fibroblast cells, it would be crucial to check if temperature aging can affect this outcome as part of their materials shelf life. Additionally, based on 24 h significant decrease in viability using Cell 165 Titer GLO assay which is based on ATP quantification, and not fully revealing the cytotoxicity of these surfaces, it would be crucial to couple this biological assessment with at least another assay like LDH cytotoxicity to evaluate the presence any cell damage because of this exposure.
Thank you very much for your suggestions. Indeed, we reported in the manuscript that the possible presence of unreacted activated oxygen species, such as hydrogen peroxides, on the material surface may have a detrimental effect on the cell viability of fibroblast, and that further studies are necessary for gaining a deeper insight in this negative effect. Then, in a future work we can use the approach you proposed for gaining more information about the cells interaction and actual cytotoxicity, a parameter of paramount importance for the proposed application. At present, this kind of study is out of the scope of the submitted paper, since both materials properties and EB treatment should be optimised before a real application could be conceived.

Round 2
Reviewer 1 Report
Dear authors, all suggested correction was done and manuscript is suitable for publication in Polymers Journal